# Association between Chewing Difficulty and Dietary Ca, Vitamin D, and Mg Intake in Korean Older Adults: 8th Korea National Health and Nutrition Examination Survey (KNHANES) (2020–2021)

**DOI:** 10.3390/nu15234983

**Published:** 2023-12-01

**Authors:** Sang-Dol Kim

**Affiliations:** Department of Nursing, College of Health Science, Kangwon National University, 346 Hwangjo-gil, Dogye-eup, Samcheok-si 25949, Gangwon-do, Republic of Korea; nu11110@kangwon.ac.kr

**Keywords:** chewing difficulty, calcium, dietary, magnesium, older adults, vitamin D

## Abstract

Nutrition intake plays a pivotal role in chewing difficulty (CD). This cross-sectional descriptive study aims to explore the associations between CD and the dietary intake of calcium (Ca), vitamin D, and magnesium (Mg) in adults aged 65 and older, utilizing data from the 8th Korea National Health and Nutrition Examination Survey (2020–2021). The chewing function was assessed using a 5-point scale questionnaire that inquired about discomfort experienced during mastication. “Very uncomfortable” and “uncomfortable,” two of the five response options, were categorized as being indicative of CD. Dietary intake was assessed through 24 h dietary recall interviews, and nutrient calculations were based on the 10th revised edition of the Korean Food Composition Table. Data (N = 2942) were analyzed using descriptive statistics and multi-logistic regression analyses in a composite sample plan file. Among the 2942 subjects, groups with insufficient daily nutrient intake had significantly higher odds ratios (ORs) for CD compared to their counterparts. In men, the ORs for Ca (1.56), Mg (1.75), and the combined intake of Mg (1.64) with Ca were elevated. In women, the ORs for Ca (1.74), Mg (1.53), and the combined intake of Mg (1.43) with Ca showed similar trends. After adjusting for age, family income, and family size variables, men’s ORs for Mg (1.55) and the combined intake of Mg (1.55) with Ca remained elevated, while women’s OR for Ca (1.58), Mg (1.42), and the combined intake of Mg (1.34) with Ca remained significant. Dietary vitamin D intake did not significantly impact the OR for CD. After adjusting for natural tooth numbers, self-perceived oral health, and obesity, no significant association was found between CD and these nutrients. In conclusion, this study underscores the importance of promoting the recommended daily intake of magnesium alongside dietary calcium to address CD.

## 1. Introduction

Chewing is a fundamental requirement for achieving sufficient nutrient intake and a vital function that remains crucial throughout one’s life [1,2]. In other words, maintaining optimal chewing functionality, which influences food choices and nutritional well-being, is necessary for sustaining oral and overall health [3]. Chewing involves breaking down and blending consumed food for effective digestion and absorption [2]. Despite its significance, not everyone possesses optimal chewing ability, and this may lead to difficulties in chewing [4].

Chewing difficulty serves as a trigger for a range of health issues, including falls, fractures, dementia, hearing loss, obesity, depression, and a diminished quality of life encompassing physical, mental, psychosocial, and economic aspects. Furthermore, these health problems could exacerbate CD, establishing a detrimental cycle [1,4,5,6,7,8]. Various factors have been linked to CD, such as tooth loss, the number of teeth, dental service utilization, lower self-rated oral health scores, disadvantaged social backgrounds, aging, lower income, and cognitive function [1,6,9,10,11,12,13,14,15].

The older population group is identified as the most vulnerable to CD [1]. This vulnerability is substantiated by the aging process, as evidenced in previous studies [1,14]. The concept of oral fragility has gained prominence in the context of aging [10]. It is anticipated that the prevalence of CD among older adults will rise in proportion to the increasing population of older individuals. According to data from the Organization for Economic Cooperation and Development (OECD), the older population accounted for approximately 17.6% of the total population in 2021, and this percentage is rising annually on a global scale [16]. In particular, as of the end of April 2023, 18.4% of the Korean population is 65 years or older [17], and Korea’s aging rate over the past decade (2011–2020) was approximately twice the OECD average (4.4% vs. 2.6%), making it the highest among the 37 OECD countries [18]. This implies that aging is an inevitable and irreversible process, recognized as a causative factor for chewing difficulty, necessitating sustainable management. Consequently, CD is acknowledged as a health problem or a geriatric syndrome that calls for consistent and targeted management [1,19]. This perspective aligns with the insights gathered from a systematic literature review comprising 35 prior studies, emphasizing the importance of maintaining optimal chewing function for achieving adequate nutrient intake [20]. Previous research has indicated that consistently poor diet quality heightens the risk of tooth loss and the accumulation of oral health issues, ultimately leading to further deterioration in dietary habits and reduced consumption of nutrient-rich foods among older individuals [21]. Hence, it is evident that the relationship between chewing difficulties and food and nutrient intake warrants ongoing investigation within the older population [2,22].

Korea has incorporated CD as an outcome indicator for the National Health Promotion Plan, integrating it into a nationwide countermeasure strategy [23]. The Danish Health Authority recommends a specific nutrient intake for older individuals and nursing home residents [24]. At a personal level, managing CD involves maintaining or enhancing dental health, but it is equally crucial to advocate for consistent consumption of dietary nutrients that impact dental health positively [2,25]. Ca intake is a notable example of nutrients essential for dental health [1,25]. This significance of chewing ability in ensuring adequate nutrient intake, particularly calcium, contributing to overall well-being, is further underscored by recent research findings [22]. Ca stands as the most abundant mineral present in human teeth and bones, playing a pivotal role as an essential nutrient for various functions contributing to human health [26]. The theoretical basis for the correlation between Ca intake and tooth loss, as well as the necessity of vitamin D for calcium absorption, has been firmly established [27]. Vitamin D enhances bone density by promoting Ca absorption in the small intestine, and its activation is facilitated by Mg [24,28]. Consequently, the combination of Ca, vitamin D, and Mg creates a synergistic effect that significantly benefits dental health [24,28,29]. Hence, the recommendation is to incorporate vitamin D and Mg consumption in conjunction with Ca [26,27,29].

Based on this, it is hypothesized that consuming a combination of Ca, vitamin D, and Mg would positively impact CD. However, research on the relationship between CD and dietary intake of Ca, vitamin D, and Mg is relatively scarce. These essential nutrients, including Ca, vitamin D, and Mg, can be obtained through prescription or non-prescription drugs, as well as from various foods. Given that drug administration is not prevalent among the older population, utilizing data on the dietary intake of Ca, vitamin D, and Mg through food is considered reasonable. A previous review study highlighted that while dietary Ca and cooperative nutrient requirements may vary slightly worldwide, diverse diets should suffice to deliver adequate Ca and essential nutrients [26]. Therefore, the objective of this study was to ascertain the association between CD and the dietary intake of Ca, vitamin D, and Mg among Korean older adults.

## 2. Materials and Methods

### 2.1. Study Design

This cross-sectional descriptive study involved a secondary analysis of data extracted from the 8th Korea National Health and Nutrition Examination Survey (KNHANES) conducted from 2020 to 2021 [23]. This study was performed following the reporting guidelines for cross-sectional research studies outlined in the Strengthening the Reporting of Observational Studies in Epidemiology (STROBE) guidelines [30]. The 8th KNHANES was carried out by the Korea Disease Control and Prevention Agency (KDCA) and encompassed a household member identification survey along with three primary areas: a health survey, a health examination, and a nutritional survey. This study specifically analyzed data from the health and nutritional surveys. The health survey included components such as a household survey, a health interview survey, and a health behavior survey. On the other hand, the nutrition survey covered dietary habits, dietary supplements, nutritional knowledge, food safety, and details of food intake from the day before the survey (24 h recall interviews). The health survey was conducted using a mobile examination vehicle, while the nutrition survey involved visits to the target households. The education and economic activity sections of the health survey, along with all components of the nutrition survey, were administered through interview-based methods. The health behavior segment within the health survey utilized a self-report approach.

### 2.2. Ethical Considerations

The 8th KNHANES received approval from the Institutional Review Board of the KDCA (2018-01-03-1C-A, approved on 30 June 2020 and 2018-01-03-2C-A, approved on 31 March 2021) [23]. In the original survey, informed consent was acquired from all participants. However, for this study, which involves secondary data analysis, obtaining informed consent from participants was deemed unnecessary. The raw data for this study were sourced through legitimate procedures from the KNHANES website under the KDCA, and all personal data had been fully anonymized.

### 2.3. Participants

Data for this analysis were sourced from the 8th KNHANES (2020–2021), specifically focusing on adults aged ≥ 65 years. The participant selection was limited to this age group (n = 2942: men = 1256, women = 1686) with normal cognitive function, excluding individuals residing in facilities such as nursing homes, hospitalized patients, those with dementia, and older adults at home experiencing severe visual, hearing, or verbal impairment. Additionally, participants who did not take part in oral examinations and surveys were excluded. No restrictions were placed on the presence of existing acute or chronic diseases, given the diversity of diseases among the participants.

### 2.4. Chewing Difficulty Assessment

The chewing function was assessed using a 5-point scale questionnaire that inquired about discomfort experienced during mastication by a team of professional investigators, including nurses, nutritionists, and health science majors [23]. For chewing difficulties, participants were asked about the degree of discomfort they felt while chewing food due to issues in their mouth, such as those related to their teeth, dentures, or gums. The response options included “very uncomfortable,” “uncomfortable,” “moderate,” “comfortable,” and “very comfortable.” During the interviews, the investigators marked a corresponding box based on the participants’ answers. The instructions to the interviewers emphasized accurately capturing the respondent’s current level of chewing discomfort [23]. “Very uncomfortable” and “uncomfortable,” two of the five response options, were categorized as being indicative of chewing difficulties, while the remaining respondents were classified into the group without chewing difficulties [31]. 

### 2.5. Assessment of dietary Ca, Vitamin D, and Mg Intakes

A survey was conducted to assess the intake of nutrients such as dietary Ca, vitamin D, and Mg from foods. The survey involved visiting specific households in accordance with the KNHANES user guide [23]. Food intake was estimated using individual 24 h dietary recall interviews. Nutrient intake from food alone was calculated, excluding dietary supplements. Nutrient intake was defined as the sum of all food and nutrient intake consumed by an individual throughout the day. 

To assess nutrient intake, participants were asked, “Please tell us what you ate and how much you ate during the day yesterday.” The investigator directly recorded all foods consumed by the participant the day before in the food intake survey table. A tertiary food code name was assigned to convert food intake into nutrient intake. The tertiary food code name served as a classification value for grouping foods with similar raw materials into one food item. It was used to calculate the quantities of major source foods for each food and nutrient. For instance, food code names like “Chwinamul”, “edible aster”, and “raw vegetables” were converted to a tertiary food code name, namely “Chwinamul”.

The 10th revised edition of the Korean Food Composition Table (KFCT), used for calculating nutrient intake in the 8th KNHNES (2020–2021), is based on the ‘National Standard Food Ingredient DB 10.0’ [24,32]. This database comprises 3270 data points and 130 ingredients. It is updated every five years by the Rural Development Administration (RDA), which serves as the Korean data center for The Food and Agriculture Organization/International Network of Food Data Systems (FAO/INFOODS). The KFCT was initially published in 1970 and has undergone regular updates since then.

The 10th revision of the KFCT focuses on the utilization of food ingredient data offline, prioritizing commonly encountered foods and ingredients in real life. It encompasses 1228 food items and 42 ingredients, providing comprehensive and updated information for nutrient analysis and dietary assessments.

### 2.6. Covariates 

Gender, age, family income level, family size, number of natural teeth, and self-perceived oral health status (SPOHS) were considered as covariates [10,12,13,33]. There were two gender groups, ‘male’ and ‘female’, and two age groups, ‘65–74’ and ‘75+’. Family income levels were categorized as ‘Lower,’ ‘Lower-middle,’ ‘Middle-upper,’ and ‘Upper.’ Family size was categorized as ‘Alone’ and ‘Two or more people.’

Oral examinations to assess the number of natural teeth were conducted by four public health dentists from the KDCA, as well as from cities and provinces. The count of natural teeth in the sample subjects excluded missing teeth, impacted teeth, or implants. The total count of natural teeth was 32, including the third molars. The count of existing teeth was categorized into three groups: 0 to 10, 11 to 20, and over 21 [15].

Regarding SPOHS, participants were asked about their perception of their dental health: “When you think about yourself, how do you feel about your oral health, including your teeth and gums?” Possible responses included ‘very good,’ ‘good,’ ‘moderate,’ ‘bad,’ or ‘very bad’ [13,33].

Regarding the comorbidity of oral diseases in the 8th KNHANES, both raw data and statistics were not publicly available. This was due to the inclusion of expert opinions, along with variations in data from dental caries (2020–2021) and periodontal biopsy (2019–2021) research, conducted by different investigators (public health dentists) [34]. Therefore, the comorbidity of oral diseases was not considered as a covariate.

Obesity was defined as having a body mass index (BMI) of 25.0 kg/m^2^ or higher, calculated by dividing weight in kilograms by the square of height in meters [23].

### 2.7. Statistical Analysis 

Data analysis was performed using IBM SPSS Statistics version 28.0.1 (IBM Corp., Armonk, NY, USA).

The composite sample plan file was generated, taking into account the strata variable (KSTRATA), weight, and cluster variable (Primary Sampling Unit, PSU) in accordance with the KNAHNES (2020–2021) user guide [34]. Sampling weights were calculated to consider the complex sampling and ensure the sample’s representativeness relative to the Korean population.

Sociodemographic variables, oral health variables, and obesity were presented as frequencies and percentages. The dietary nutrient intake variables were expressed as mean ± standard error (SE) or as frequencies and percentages. The prevalence of CD was presented as percentages along with the standard error. The difference in CD according to sociodemographic variables, oral health variables, obesity, and nutrient variables was assessed using the weighted chi-square test.

For logistic analysis, dietary nutrient intakes (Ca, vitamin D, and Mg) were categorized into two groups as follows: in older men, <700 mg/day and ≥700 mg/day for Ca, <15 μg/day and ≥15 μg/day for vitamin D, and <370 mg/day and ≥370 mg/day for Mg. In older women, dietary nutrient intakes were categorized as <800 mg/day and ≥800 mg/day for Ca, <15 μg/day and ≥15 μg/day for vitamin D, and <280 mg/day and ≥280 mg/day for Mg. These specific cutoff values were selected in accordance with the 2020 Korean Dietary Reference Intakes (KDRIs) Recommended Nutrient Intakes (RNI) [35].

Weighted logistic regression was employed to examine the association between CD and sociodemographic variables (sex, age group, family income level, and family size), as well as oral health characteristics (number of natural teeth and self-perceived oral health status), and the obesity variable. Additionally, weighted multivariate logistic regression was used to investigate the association between CD and dietary nutrient intakes (Ca, vitamin D, and Mg) based on the 2020 KDRIs’ RNI. The analysis was divided into an unadjusted model and two adjusted models. The first adjusted model involved adjusting for covariates such as age, family income, and family size variables that exhibited a significant relationship with CD. The second adjusted model involved adjusting for covariates related to the number of natural teeth and SPOHS variables that had a significant relationship with CD. The obesity variable showed no significant association with CD and was consequently excluded as an adjustment variable in the logistic analysis.

The association was presented as an odds ratio (OR) and 95% confidence interval (CI). *p*-values < 0.05 were considered statistically significant.

## 3. Results

### 3.1. Chewing Difficulty According to Sociodemographic, Oral Health Characteristics, and Obesity

The participant characteristics and chewing difficulty according to sociodemographic factors, oral health characteristics, and obesity are outlined in Table 1. The average age of the participants was 73.0 years. The total number of participants was 2942, with 1256 (45.7%) being men and 1686 (53.3%) being women. Approximately 33.7% of the participants self-reported difficulty in chewing. Significant differences in CD were observed based on sex, age, family income, family size, the number of natural teeth, and self-reported oral health between the two groups, respectively (*p* < 0.05). However, there were no significant differences in CD according to obesity (*p* > 0.05).

### 3.2. Chewing Difficulty According to Dietary Nutrient Intakes

Chewing difficulty related to dietary nutrient intakes among Korean older adults in 2020–2021, based on the 2020 KDRIs’ RNI, is presented in Table 2. There were significant differences in CD based on dietary Ca and Mg intakes in both the men’s group (*p* < 0.05) and the women’s group (*p* < 0.05). However, there were no significant differences in CD based on dietary vitamin D intakes in either the men’s or the women’s groups (*p* > 0.05).

### 3.3. Association between Chewing Difficulty and Sociodemographic and Oral Health Characteristics

The association between CD and sociodemographic variables, the number of natural teeth, SPOHS, and obesity is presented in Table 3. 

There were associations with a higher OR for CD in various groups as follows: women (OR: 1.39, 95% CI: 1.16–1.66, *p* < 0.001), individuals aged 75 years and over (OR: 1.66; 95% CI: 1.39–2.00, *p* < 0.001), those in the lower family income group (OR: 2.40; 95% CI: 1.69–3.41, *p* < 0.001), people living alone (OR: 1.32, 95% CI: 1.08–1.60, *p* < 0.05), individuals with 1–10 teeth (OR: 2.65, 95% CI: 2.10–3.34, *p* < 0.001), and those with a very low score on SPOHS (OR: 49.3, 95% CI: 6.86–354.36, *p* < 0.001) when compared to the reference group, respectively. However, there was no significant association between obesity and CD (OR: 0.87, 95% CI: 0.72–1.05, *p* > 0.05). 

### 3.4. Association between Chewing Difficulty and Dietary Nutrients Intakes

The association between CD and the dietary nutrient intakes according to the 2020 KDRIs’ RNI is presented in Table 4. 

For the group of older men, a dietary Ca intake of less than 700 mg/day (OR: 1.56, 95% CI: 1.13–2.15, *p* < 0.05) and a dietary Mg intake of less than 370 mg/day (OR: 1.75, 95% CI: 1.31–2.36, *p* < 0.001) were significantly associated with a higher OR for CD when compared to the reference group. After adjusting for age, family income level, and family size, a dietary Mg intake of less than 370 mg/day (OR: 1.55, 95% CI: 1.14–2.09, *p* < 0.05) remained significantly associated with a higher OR for CD when compared to the reference group. However, when adjusting for the number of natural teeth and the SPOHS score, a dietary Ca intake of less than 700 mg/day (OR: 1.34, 95% CI: 0.89–2.02, *p* > 0.05) and a dietary Mg intake of less than 370 mg/day (OR: 1.40, 95% CI: 0.96–2.04, *p* > 0.05) were not significantly associated with a higher OR for CD when compared to the reference group. A dietary Mg intake of less than 370 mg/day (OR: 1.63, 95% CI: 1.17–2.26, *p* < 0.05) was significantly associated with a higher OR for CD when coexisting with a dietary Ca intake of less than 700 mg/day, as compared to the reference group. After adjusting for age, family income level, and family size, a dietary Mg intake of less than 370 mg/day (OR: 1.55, 95% CI: 1.12–2.14, *p* < 0.05), when combined with a dietary Ca intake of less than 700 mg/day, also remained significantly associated with a higher OR for CD, as compared to the reference group.

For the group of older women, a dietary Ca intake of less than 800 mg/day (OR: 1.74, 95% CI: 1.19–2.54, *p* < 0.05) and a dietary Mg intake of less than 280 mg/day (OR: 1.53, 95% CI: 1.21–1.95, *p* < 0.001) were significantly associated with a higher OR for CD when compared to the reference group, respectively. After adjusting for age, family income level, and family size, a dietary Ca intake of less than 800 mg/day (OR: 1.58, 95% CI: 1.07–2.33, *p* < 0.05) and a dietary Mg intake of less than 280 mg/day (OR: 1.42, 95% CI: 1.12–1.80, *p* < 0.05) remained significantly associated with a higher OR for CD when compared to the reference group. However, when adjusting for the number of natural teeth and the SPOHS score, a dietary Ca intake of less than 800 mg/day (OR: 1.20, 95% CI: 0.73–1.99, *p* > 0.05) and a dietary Mg intake of less than 280 mg/day (OR: 1.26, 95% CI: 0.95–1.66, *p* > 0.05) were not significantly associated with a higher OR for CD when compared to the reference group. A dietary Mg intake of less than 280 mg/day (OR: 1.43, 95% CI: 1.11–1.84, *p* < 0.05), in combination with a dietary Ca intake of less than 800 mg/day, was significantly associated with a higher OR for CD when compared to the reference group. After adjusting for age, family income level, and family size, a dietary Mg intake of less than 280 mg/day (OR: 1.34, 95% CI: 1.04–1.72, *p* < 0.05), in combination with a dietary Ca intake of less than 800 mg/day, also remained significantly associated with a higher OR for CD when compared to the reference group.

## 4. Discussion

This cross-sectional descriptive study involved a secondary analysis using data from the 8th Korea National Health and Nutrition Examination Survey (2020–2021), targeting seniors aged 65 years or older. It confirmed the association between the prevalence of CD and the intake of Ca, vitamin D, and Mg from foods.

The prevalence of CD among Korean older adults was 33.7%. This finding closely aligns with the pooled prevalence of CD (35%) reported among older individuals in long-term care, as indicated in a meta-analysis of 12 previous studies [1]. The studies included in the meta-analysis were published between 1998 and 2020, and they were conducted in various countries, including Brazil (six studies), as well as the United States, Germany, Norway, New Zealand, Great Britain, and South Korea (one study each). According to the meta-analysis, CD’s prevalence among older adults in long-term care facilities was one in three [1]. Based on the data from this meta-analysis, it can be estimated that the prevalence of CD in the older population has remained relatively consistent across countries over the past 20 years.

As shown in these findings, sociodemographic variables significantly associated with the CD include sex, age, family income level, and family size. The association between sex and CD reveals that older women are linked to a higher OR for CD than older men. These findings are consistent with studies conducted in the Netherlands, Poland, and the six countries included in the meta-analysis [1,12,36]. These findings can be interpreted as indicating that older women are vulnerable to CD, emphasizing the need to promote dietary consumption and nutrient absorption in older women for the alleviation of CD, along with the elimination of barriers and continuous practical education [1,12,36]. 

Older age groups were associated with a higher OR for CD, which is consistent with findings observed in the oldest age groups in Taiwan and the Netherlands, where there was a higher prevalence of CD [13,36]. This can be attributed to the fact that aging is the main contributing factor to the prevalence of CD [1,14]. Therefore, it is necessary to maintain dental health and ensure sufficient dietary nutrient intake from a young age rather than waiting until the onset of aging. 

When considering family income levels, the OR for experiencing CD tended to be higher in the low-income group. These findings align with a prior study that demonstrated a higher likelihood of CD in low-income Brazilian adults compared to their high-income counterparts [33]. These findings carry broader implications for public health and lend support to arguments against stringent economic austerity measures. In Korea, the cost of dental treatment is notably high as it is not covered by insurance. Furthermore, the poverty rate among the older population in Korea ranks as the highest among OECD countries. Many older individuals are hesitant to seek dental care due to the substantial financial burden associated with dental examinations and treatment, which is exacerbated by the lack of income-generating activities. This underscores the necessity for providing intensive support to economically vulnerable older populations. 

When examining family size, the OR for experiencing CD was higher among individuals living alone compared to those with the presence of family members. These findings align with a previous study that identified family members as a significant covariate associated with the prevalence of CD in Brazilian adults [33]. It is believed that this covariate influences the intake of essential nutrients through the diet, possibly because the tradition of communal family meals has gradually declined. This supports the findings of prior research that eating alone is linked to poorer dental conditions [10]. 

Additionally, in this study, the OR for experiencing CD tended to be higher in the group with a lower number of natural teeth. These findings are self-evident, as teeth are crucial for biting and chewing food. This corresponds with previous studies that have demonstrated a strong association between the number of teeth and daily Ca intake in older individuals [3,10,15,21]. Not surprisingly, a consistently poor diet quality is linked to an increased risk of tooth loss and the accumulation of oral health problems [3,21]. Therefore, ensuring optimal chewing function to achieve adequate nutrient intake necessitates management at both the individual and national levels [1,15,21,37]. Moreover, the OR for experiencing CD tended to be significantly higher in the group with lower SPOHS scores. These findings align with previous studies suggesting that self-perceived oral health might be influenced by CD [38,39]. It was confirmed that poor subjective oral health status had a more pronounced impact on chewing function, and the prevalence of CD was lower when dietary calcium was consumed. Therefore, it can be inferred that there is a need to promote dietary calcium intake to maintain optimal chewing function. However, there was no significant relationship between obesity and CD. In contrast to the findings of this study, it has been reported that CD can influence eating speed and portion size, potentially leading to obesity [40]. This highlights the need for further longitudinal investigations into the relationship between CD and obesity in the elderly [6]. 

On the other hand, it is a well-known fact that CD is closely related to nutrient intake. According to the 2020 Korean Dietary Reference Intakes, the recommended dietary nutrient intakes for men aged 65 and over are as follows: 700 mg/day for Ca, 15 μg/day for vitamin D, and 370 mg/day for Mg [35]. For women aged 65 and over, the recommended dietary nutrient intakes are 800 mg/day for Ca, 15 μg/day for vitamin D, and 280 mg/day for Mg [35]. As evident from the findings of this study, the OR for CD was significantly higher in groups of older men and women with dietary Ca and Mg intakes below the recommended daily amounts specified by the 2020 KDRIs, with the exception of vitamin D. These findings are presumed to be a result of unbalanced dietary habits among older Korean adults [2,15,25]. The primary reason for Ca nutrient deficiency was the lower frequency of milk and dairy product consumption among the Korean population aged 65 and over when compared to other age groups [25]. Despite the Danish Health Authority’s recommendation that all residents of Denmark’s nursing homes should consume daily supplements of 800–1000 mg of Ca and 20 μg of vitamin D, these recommendations were poorly implemented [24]. This directly affects the absorption of Ca nutrients from the small intestine, leading to an increased incidence of fractures and falls, as well as difficulty with chewing [24]. A leading study pointed out that homebound older adults have very insufficient intake of Ca, vitamin D, and Mg, which are nutrients required for the musculoskeletal system and are mainly obtained through dietary preparation and consumption [39]. From previous studies and the present study, it is evident that older adults continue to display inadequate intake of Ca, vitamin D, and Mg [24,25,41]. This holds true for both dietary nutrients and supplement nutrients. Furthermore, previous studies have emphasized the necessity of implementing measures to overcome barriers to nutrient intake [24,25,41].

The present study hypothesized that the combined intake of dietary Ca, vitamin D, and Mg would be associated with a lower OR for CD in Korean older adults [1,5,24,41]. However, as dietary vitamin D intake did not have an effect on the OR of CD, the analysis considered the combined intake of dietary Ca and Mg when calculating the OR for CD. In the present study, when adjusting for age, family income level, and family size variables, the OR for CD was significantly higher in both older men and women groups who consumed less dietary Mg than the recommended daily amount. In the case of dietary Ca intake, it was only significant in the older women’s group. However, adjusting for the natural number of teeth and SPOHS variables did not significantly affect the OR of CD in the relationship between dietary Ca, vitamin D, Mg intake and CD in both older men and women groups. Meanwhile, when the combined intake of dietary Ca and Mg is considered, the OR for experiencing CD was higher in the older adult groups who consumed less dietary Mg than the recommended daily amount, whether unadjusted or adjusted for age, family income level, and family size variables. In contrast, adjusting for natural tooth numbers and SPOHS variables did not show an association with the OR for CD in the older adult groups who consumed less dietary CD or dietary Mg than the recommended daily amount. Previous studies have suggested that vitamin D intake enhances Ca absorption, while the combined intake of Ca, vitamin D, and Mg improves bone density, and the combined dietary intake of Ca and Mg reduces hearing loss in older adults [5,27,29]. In contrast to the views presented in previous studies [27,29], the current findings indicate that dietary Ca intake, dietary Mg intake, and the combined intake of dietary Ca and Mg, rather than dietary vitamin D intake, are significantly associated with a higher OR for CD in older Korean adults. This suggests that the dietary intake of Ca and Mg in Korean older adults is considerably lower in comparison to the RNI established by the KDRIs in 2020. Surprisingly, according to the findings of this study, dietary Mg intake, when consumed concurrently with dietary Ca, is a nutrient that increases the OR of CD in elderly people, irrespective of whether the subject’s general characteristics or oral health variables are adjusted. This suggests that to reduce the prevalence of CD in older adults, it is important to encourage dietary Mg intake while also consuming dietary Ca. 

As it turns out, Mg acts as a Ca antagonist, inhibiting Ca absorption in the small intestine. Ca is a vital component that directly impacts bone health, and it is a nutrient that requires active intake because our bodies do not produce it. According to the KDRIs in 2020 announced by the Ministry of Health and Welfare, Ca intake is insufficient in the elderly, who experience rapid bone density reduction [35]. In particular, the consumption of instant foods and soft drinks exacerbates Ca deficiency. Mg is also one of the nutrients lacking among the elderly in Korea. Mg, often referred to as a “natural sedative,” is another nutrient that our bodies cannot produce, so deficiency symptoms can easily occur if we do not maintain a balanced diet. Surprisingly, Ca and Mg, both of which are often lacking, have a synergistic effect when consumed together. In other words, most of the Ca in the body is located in the teeth and bones, and if Ca intake is insufficient, our bodies extract Ca from the bones for use. In this process, Mg plays a role in Ca metabolism and acts as an antagonist to facilitate the release of stored Ca. 

Paradoxically, CD becomes a leading factor that results in poor nutritional status, which, in turn, leads to CD due to the lack of essential nutrients for dental health [15]. It is crucial to maintain optimal chewing function to ensure sufficient nutrient intake, including Ca, vitamin D, and Mg, which are vital for overall well-being [20,22,42]. In particular, dietary Ca has been reported to offer topical enamel protection, which can prevent tooth loss due to cavities [15]. Based on the findings of this study, it is also important not to ignore the recommended daily intake of Mg to optimize Ca absorption. Contrary to the findings of this study, it is widely recognized that Ca, vitamin D, and Mg typically exhibit a synergistic effect when consumed together. Therefore, this underscores the necessity of developing methods and strategies for the recommended daily nutrient intake in older people with CD. In other words, this means that continuous practical education should be provided to target the elimination of barriers and promote the recommended daily nutrient intake [12]. Furthermore, new food products should be developed to encourage healthy eating patterns, with a focus on fresh foods rather than instant foods, for older adults with chewing difficulties [2,12,43,44]. 

Taken together, this suggests that, in order to reduce the prevalence of CD in older adults, it is important to promote dietary Mg intake while also ensuring adequate dietary Ca consumption. Additionally, the variables that influenced the OR of CD in older Korean adults included age, family income level, and family size, while SPOHS, the number of natural teeth, and obesity did not have a significant impact. The SPOHS variable can be influenced by various factors, whereas natural teeth are irreplaceable but can be supplemented with dentures or prosthetics. Therefore, this implies that factors such as age, family income level, and family size should be considered when addressing the prevalence of CD in older adults.

The present study has several limitations. Firstly, the data in the 8th KNHNES were obtained through self-reported interviews. Self-reported surveys are generally considered to have less validity and reliability compared to objective measures. There remains a debate about the accuracy of self-reported CD in older adults. Fortunately, a valid assessment tool for measuring chewing problems in the Korean older population has recently been developed [45]. Further investigations using this tool will be a significant step in addressing the issue with the accumulation of unbiased, reliable, and valid data on chewing difficulties in older adults.

Moreover, the present study examined and analyzed chewing difficulties regardless of whether participants had undergone dental check-ups, had oral health or teeth problems, and consumed Ca, vitamin D, and Mg supplements through non-prescription or prescription medications. This likely introduced bias into the assessment of chewing difficulties, as the history of dental or other therapies for oral or general health problems may vary among older adults. This suggests that healthcare history should be considered alongside the assessment of chewing difficulties in older adults.

Additionally, this study only adjusted for sociodemographic factors, oral health characteristics, including the number of natural teeth and self-perceived oral health status, and obesity. Factors such as comorbidities, general health status, and the number of teeth are relevant to oral health and nutrition, especially in older adults. Therefore, the author recommends further investigations into the association between chewing difficulties and variables such as comorbidities, general health status, and the number of teeth. 

This study was only able to determine the relationship between food nutrient intake and the prevalence of CD over a specific period of time due to its cross-sectional design. Therefore, this study proposes a follow-up investigation based on a longitudinal study design to more comprehensively assess the association of CD prevalence with food intake over time. Finally, the present study has limitations in terms of its generalizability at the national level, as it only included the older population in Korea.

## 5. Conclusions

There was a significant association with a higher OR for CD in groups of Korean older adults who consumed less than the recommended daily amounts of dietary Ca and Mg according to the 2020 Korean Dietary Reference Intakes. Specifically, there was a significant association with a higher OR for CD in the groups where the combined intake of dietary Mg and Ca was less than the recommended daily amounts. Age, family income, and family size served as significant adjustment variables, while natural tooth numbers, self-perceived oral health status, and obesity variables did not. This highlights the importance of promoting the consumption of the recommended amount of dietary Mg in addition to dietary Ca.

## Figures and Tables

**Table 1 nutrients-15-04983-t001:** Chewing difficulty according to sociodemographic factors, oral health characteristics, and obesity in Korean older adults (2020–2021).

Characteristics	Unweighted N (%)	Chewing Difficulty % (SE)		*p* Value
		No	Yes	
Sex				
Male	1256 (45.7)	70.2 (1.6)	29.8 (1.6)	0.001
Female	1686 (54.3)	62.9 (1.4)	37.1 (1.4)	
Total	2942 (100.0)	66.3 (1.1)	33.7 (1.1)	
Age group (years), Mean ± SE	73.0 ± 0.1	72.6 ± 0.2	73.9 ± 0.2	0.000
65–74	1684 (59.0)	40.6 (1.7)	59.4(1.7)	
75+	1258 (41.0)	29.1 (1.4)	70.9 (1.4)	
Total	2942 (100.0)	33.7 (1.1)	66.3 (1.1)	
Family income level				
Lower	1338 (42.3)	60.4 (1.7)	39.6 (1.7)	0.000
Lower middle	841 (29.4)	65.6 (2.0)	34.4 (2.0)	
Middle upper	454 (17.0)	73.6 (2.3)	26.4 (2.3)	
Upper	273 (11.3)	78.6 (2.8)	21.4 (2.8)	
Total	2916 (100.0)	66.3 (1.1)	33.7 (1.1)	
Family size				
Alone	782 (21.6)	61.4 (1.9)	38.6 (1.9)	0.006
Two or more people	2160 (78.4)	63.6 (1.3)	32.4 (1.3)	
Total	2942 (100.0)	66.3 (1.1)	33.7 (1.1)	
Number of natural teeth, Mean ± SE	17.9 ± 0.3	19.5 ± 0.3	15.1 ± 0.4	0.000
1–10 teeth	610 (24.5)	54.3 (2.4)	45.7 (2.4)	
11–20 teeth	508 (22.4)	59.1 (2.7)	40.9 (2.7)	
20+ teeth	1201 (53.1)	75.9 (1.5)	24.1 (1.5)	
Total	2319 (100.0)	66.9 (1.3)	33.1 (1.3)	
Self-perceived oral health status				
Very good	23 (1.0)	96.5 (3.4)	3.5 (3.4)	0.000
Good	388 (17.4)	86.7 (2.0)	13.3 (2.0)	
Moderate	960 (41.1)	75.6 (1.8)	24.4 (1.8)	
Bad	835 (35.3)	50.8 (2.0)	49.2 (2.0)	
Very Bad	113 (5.2)	36.0 (5.1)	64.0 (5.1)	
Total	2319 (100.0)	66.9 (1.3)	33.1 (1.3)	
Obesity				
No (BMI ≤ 25 kg/m^2^)	1038 (37.6)	31.3 (1.7)	68.7 (1.7)	0.149
Yes (BMI > 25 kg/m^2^)	1733 (62.4)	34.4 (1.5)	65.6 (1.5)	
Total	2771 (100.0)	33.3 (1.2)	66.7 (1.2)	

BMI; body mass index, N; number, SE; standard error.

**Table 2 nutrients-15-04983-t002:** Chewing difficulty related to dietary nutrient intakes among Korean older adults in 2020–2021, based on the 2020 KDRIs’ RNI.

Nutrients	Unweighted N (%)	Chewing Difficulty		*p* Value
		% (SE)		
		No	Yes	
Men				
Ca, Mean ± SE	517.4 ± 10.1	545.4 ± 12.5	457.8 ± 15.3	
<700 mg/day	987 (78.2)	68.2 (1.7)	31.8 (1.7)	0.006
≥700 mg/day	254 (21.8)	77.0 (2.7)	23.0 (2.7)	
Total	1241 (100.0)	70.1 (1.6)	29.9 (1.6)	
VITD, Mean ± SE	3.03 ± 0.2	3.3 ± 0.2	2.5 ± 0.3	
<15 μg/day	1209 (96.6)	69.8 (1.6)	30.2 (1.6)	0.205
≥15 μg/day	32 (3.4)	80.9 (7.9)	19.1 (7.9)	
Total	1241 (100.0)	70.1 (1.6)	29.9 (1.6)	
Mg, Mean ± SE	337.4 ± 5.3	353.6 ± 6.2	302.4 ± 8.1	
<370 mg/day	818 (65.5)	66.3 (2.0)	33.7 (2.0)	0.000
≥370 mg/day	423 (34.5)	77.5 (2.2)	22.5 (2.2)	
Total	1241 (100.0)	70.1 (1.6)	29.9 (1.6)	
Women				
Ca, Mean ± SE	427.7 ± 9.2	452.8 ± 11.4	393.3 ± 12.9	
<800 mg/day	1480 (90.3)	61.8 (1.5)	38.2 (1.5)	0.003
≥800 mg/day	157 (9.7)	73.7 (3.7)	26.3 (3.7)	
Total	1637 (100.0)	62.9 (1.4)	37.1 (1.4)	
VITD, Mean ± SE	2.2 ± 0.1	2.3 ± 0.1	2.06 ± 0.3	
<15 μg/day	1615 (98.6)	63.1 (1.4)	36.9 (1.4)	0.313
≥15 μg/day	22 (1.4)	50.6 (12.5)	49.4 (12.5)	
Total	1637 (100.0)	62.9 (1.4)	37.1 (1.4)	
Mg, Mean ± SE	268.0 ± 4.4	277.2 ± 5.1	256.0 ± 7.2	
<280 mg/day	994 (60.4)	59.1 (1.7)	40.9 (1.7)	0.000
≥280 mg/day	644 (39.6)	68.8 (2.2)	31.2 (2.2)	
Total	1637 (100.0)	62.9 (1.4)	37.1 (1.4)	

Ca; calcium, KDRIs; Korean dietary reference intakes, Mg; magnesium, N; number, RNI; recommended nutrient intake, SE; standard error, VITD; vitamin D.

**Table 3 nutrients-15-04983-t003:** Association between chewing difficulty and sociodemographic and oral health characteristics in Korean older adults (2020–2021).

Characteristics	Weighted OR	95% CI	*p* Value
		Lower-Upper	
Sex			
Female	1.39	1.16–1.66	0.000
Male	1	Reference	
Age group (years)			
75+	1.66	1.39–2.00	0.000
65–69	1	Reference	
Family income level			
Lower	2.40	1.69–3.41	0.000
Lower middle	1.92	1.34–2.77	0.000
Middle upper	1.32	0.88–1.97	0.176
Upper	1	Reference	
Family size			
Alone	1.32	1.08–1.60	0.006
Two or more people	1	Reference	
Number of natural teeth			
1–10 teeth	2.65	2.10–3.34	0.000
11–20 teeth	2.18	1.66–2.86	0.000
20+ teeth	1	Reference	
Self-perceived oral health status			
Very bad	49.3	6.86–354.36	0.000
Bad	26.79	3.68–194.91	0.001
Moderate	8.94	1.25–63.92	0.029
Good	4.23	0.58–31.14	0.156
Very good	1	Reference	
Obesity			
No (BMI ≤ 25 kg/m^2^)	0.87	0.72–1.05	0.150
Yes (BMI > 25 kg/m^2^)	1	Reference	

BMI; body mass index, CI; confidence interval, OR; odds ratio.

**Table 4 nutrients-15-04983-t004:** Association between chewing difficulty and dietary nutrient intakes according to the 2020 KDRIs’ RNI in Korean older adults (2020–2021).

Nutrients	Unadjusted		Age, Family Income, and Family Size Adjusted		Number of Natural Teeth and SPOHS Adjusted	
	Weighted OR	95% CI	Weighted OR	95% CI	Weighted OR	95% CI
		Lower-Upper		Lower-Upper		Lower-Upper
Men						
Ca						
<700 mg/day	1.56 *	1.13–2.15	1.38	0.99–1.94	1.34	0.89–2.02
≥700 mg/day	1	Reference	1	Reference	1	Reference
Vitamin D						
<15 μg/day	1.84	0.68–4.98	1.58	0.60–4.13	1.79	0.48–6.62
≥15 μg/day	1	Reference	1	Reference	1	Reference
Mg						
<370 mg/day	1.75 **	1.31–2.36	1.55 *	1.14–2.09	1.40	0.96–2.04
≥370 mg/day	1	Reference	1	Reference	1	Reference
Combined intake						
Ca						
<700 mg/day	1.23	0.86–1.77	1.00	0.63–1.58	1.17	0.75–1.81
≥700 mg/day	1	Reference	1	Reference	1	Reference
Mg						
<370 mg/day	1.63 *	1.17–2.26	1.55 *	1.12–2.14	1.33	0.89–1.98
≥370 mg/day	1	Reference	1	Reference	1	Reference
Women						
Ca						
<800 mg/day	1.74 *	1.19–2.54	1.58 *	1.07–2.33	1.20	0.73–1.99
≥800 mg/day	1	Reference	1	Reference	1	Reference
Vitamin D						
<15 μg/day	0.60	0.22–1.61	0.58	0.21–1.61	0.44	0.14–1.38
≥15 μg/day	1	Reference	1	Reference	1	Reference
Mg						
<280 mg/day	1.53 *	1.21–1.95	1.42 *	1.12–1.80	1.26	0.95–1.66
≥280 mg/day	1	Reference	1	Reference	1	Reference
Combined intake						
Ca						
<800 mg/day	1.41	0.94–2.11	1.34	0.89–2.01	1.06	0.62–1.82
≥800 mg/day	1	Reference	1	Reference	1	Reference
Mg						
<280 mg/day	1.43 *	1.11–1.84	1.34 *	1.04–1.72	1.24	0.92–1.67
≥280 mg/day	1	Reference	1	Reference	1	Reference

Ca; calcium, CI; confidence interval, KDRIs; Korean dietary reference intakes, Mg; magnesium, OR; odds ratio, RNI; recommended nutrient intake. *; *p* < 0.5, **; *p* < 0.01.

## Data Availability

There are restrictions on the availability of these data. Data were obtained from the Korea Disease Control and Prevention Agency and are available at https://knhanes.kdca.go.kr/knhanes/sub03/sub03_02_05.do with the permission of the KDCA (accessed on 10 April 2023).

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
