# Peer review of "Association between Chewing Difficulty and Dietary Ca, Vitamin D, and Mg Intake in Korean Older Adults: 8th Korea National Health and Nutrition Examination Survey (KNHANES) (2020–2021)"

_nutrients, 2023, doi:10.3390/nu15234983_

Round 1

Reviewer 1 Report (Previous Reviewer 1)

Comments and Suggestions for Authors

This is an interesting study that addresses the association between chewing difficulty and the intake of several Mindy using a National representative data from south Korea. The study is valid however some points should be addressed to enhance the manuscript.

---

Abstract:

Improve clarity on chewing difficulty measurement. Report the methodology briefly in the abstract.

Method:

Chewing Difficulty Assessment:

Clarification is needed on how responses to the two questions form the chewing difficulty variable. Provide detailed elaboration.

Statistical Analysis:

Detail the adjustment strategy for each logistic regression model.

Results:

Table 4:

Confirm if the last model is “additionally” adjusted for oral health variables.

Tables 3 and 4:

Replace "referent" with "reference" for consistency and accuracy.

Discussion:

Begin with a concise summary of the study's objectives and significant findings. Address the limitation of using cross-sectional data which affects the determination of temporality.

Author Response

Dear reviewer,

Reviewer 2 Report (New Reviewer)

Comments and Suggestions for Authors

Dear Author,

Nutrition intake plays a pivotal role in chewing difficulty (CD). This cross-sectional study 8 aims to explore the associations between CD and dietary calcium (Ca), vitamin D, and magnesium 9 (Mg) intake in adults aged 65 and older, utilizing data from the 8th Korea National Health and 10 Nutrition Examination Survey (2020–2021).

The study is of scientific interest and in line with the aims of the Journal. However, there are some issues that should be added.

Abstract and Methods Sections were well organized.

·      Was The study was conducted according to the “Strengthening the Reporting of Observational Studies in Epidemiology” (STROBE)?

·      Line 121. “(n = 2,942: men= 1,256, women = 1,686)”, Please report this information in the Result section.

 Results

The result Section was clear.

Discussion

Discussion was well organized. 

Author Response

Dear reviewer,

This manuscript is a resubmission of an earlier submission. The following is a list of the peer review reports and author responses from that submission.

Round 1

Reviewer 1 Report

Comments and Suggestions for Authors

This is an interesting study that address the association between chewing difficulty and nutritional intake using a national representative sample from South Korea. However, there are some points that should be addressed by the authors to enhance the overall quality of the manuscript.

Abstract:

Please include details on how chewing difficulty was measured in the study. Additionally, provide more information about dietary intake in the abstract to provide readers with context about the exposure and outcome of the study.

Briefly describe the statistical method used in the study.

Provide a clear conclusion of the study.

Introduction:

Include detailed information on the current understanding of the relationship between oral health and nutritional intake. As of now, the evidence is inconclusive due to variations in study designs and measurements of oral health indicators (including chewing difficulties) and nutritional outcomes (such as non-nutritional intake).

Methods:

Provide more information on what was considered as dietary calcium (Ca), vitamin D, and magnesium (Mg) intake. For example, clarify whether this exclusively referred to raw food or included dietary supplements.

Include a section on the covariates that were tested, how they were obtained and managed, and why they were specifically chosen.

One major concern with the statistical analysis is the lack of accounting for important factors that could impact the association, such as comorbidity and general health status, and number of teeth. These factors are relevant for oral health and nutrition, especially in older adults. Consider redoing the analysis and adjusting for these factors.

Results:

Since the aim of the study is to identify the association between chewing difficulty (CD) and dietary calcium, vitamin D, and magnesium intakes among Korean older adults, there is no need to interpret the odds ratio of the covariates. Refrain from interpreting the covariates in Table 2. For more information, refer to the article by Westreich, D. and Greenland, S. (2013), titled "The table 2 fallacy: presenting and interpreting confounder and modifier coefficients" in the American Journal of Epidemiology, 177(4), pp. 292-298.

Revise Table 2 to make it less confusing. Include the unadjusted and fully adjusted models.

Author Response

Dear reviewer, thank you for your feedback. I have carefully taken your comments into consideration and applied the necessary edits directly onto the manuscript. The revisions are colored in red. The corresponding responses to the feedback can be found in in the attached file.

Reviewer 2 Report

Comments and Suggestions for Authors
This study investigated the associations between chewing difficulty and dietary calcium, vitamin D, and magnesium intake in older Korean adults. The data analysed was obtained from the 8th Korea National Health and Nutrition Examination Survey conducted between 2020 and 2021, involving 2,942 adults aged 65 years and above. The prevalence of CD was found to be 33.7%. The results showed that dietary Ca intake and total Ca intake, adjusted for confounding factors such as sex, age, family income, and family members, were associated with lower odds of CD prevalence. Based on these findings, the authors suggest implementing strategies to improve dietary Ca intake in older adults to address the issue of CD. Overall, this study highlights the relationship between CD and dietary Ca intake in older Korean adults and emphasises the importance of nutritional management to ensure this population's tolerable intake of dietary nutrients.

The health status of the participants in the mentioned study is not clearly defined, which may have implications for interpreting the findings. To understand the relationship between chewing ability, nutrient intake (including calcium), and overall health in older adults, it is crucial to consider factors such as energy intake, protein consumption, BMI, and malnutrition. These factors can significantly impact muscle function, chewing capacity, and nutritional status. Similarly, cognitive function should also be considered, as it can influence dietary choices and nutrient intake.

Inclusion and exclusion criteria were not mentioned.

What software was used for calculating the nutrient intakes?

To further support the importance of maintaining good chewing function for adequate nutrient intake, including calcium, I recommend referring to the following studies: 
"Relationship between chewing ability and nutritional status in Japanese older adults: A cross-sectional study" DOI: 10.3390/ijerph18031216. This study explores the associations between chewing ability, nutritional status, and quality of life in older adults. It demonstrates that individuals with better chewing ability exhibit higher calcium and protein intakes and improved nutritional status. These findings emphasise the significance of chewing ability in ensuring sufficient nutrient intake, including calcium, which contributes to overall well-being. "Systematic review of the association of mastication with food and nutrient intake in the independent elderly" DOI: 10.1016/j.archger.2014.08.005. Considering these studies, you will gain valuable insights into the relationship between chewing ability, nutrient (calcium) intake, and their broader implications for older adults' health. These findings underscore the importance of maintaining optimal chewing function for achieving adequate nutrient intake, including calcium, vital for overall well-being.

Author Response

(The authors gave the same response as above.)

Reviewer 3 Report

Comments and Suggestions for Authors

This study aims to the associations between chewing difficulty and dietary Ca, vitamin D, and Mg intakes in Korean older adults. The manuscript has several issues that need to be addressed.

1. The authors should clarify the design of the study in the title.

2. Please provide reference for the definition of chewing difficultly in this study (line 96-103).

3. It is unclear if the nutrient calculation method is valid. Pease provide a reference that could prove the validity of the nutrient calculation method (line 108-118). Showing the unit of each variable here is also preferred.

4. Please provide more detailed information about the logistic model in the Statistical analysis. For example, specifying the independent and dependent variable, interaction terms, the confounding factors in each model. This information will help the readers understand the results in Table 2.

Indeed, I found it is difficult to understand Table 2 and corresponding text. Is it correct that the estimates of intercept of each model was used as the OR for dietary Ca intake? How the author modeled the variable dietary Ca intake in the analysis (specifically, “when dietary Ca intake is reduced by 1mg/day” in line 157-158)?

There is no statistics for the interaction terms in Table 2.

Please also provide the full names for each abbreviation (as footnote) in Table.

5. Line 200-202: “These findings were consistent 200 with those of studies conducted in the Netherlands, Poland, and the six countries included 201 in the meta-analysis [3, 19, 20]”. Did studies of No. 3.19.20 also examined the associations between Ca intake and chewing difficulty?

Of the studies of No. 20 and 21, which study included sample from Taiwan?

Based the above issues, I would suggest the author double-check each reference to see if all the references are appropriate.

Author Response

(The authors gave the same response as above.)

Round 2

Reviewer 1 Report

Comments and Suggestions for Authors

Thanks for addressing my comments, however, one of my comments has not been fully addressed 
 "One major concern with the statistical analysis is the lack of accounting for important factors that could impact the association, such as comorbidity and general health status, and number of teeth. These factors are relevant for oral health and nutrition, especially in older adults. Consider redoing the analysis and adjusting for these factors."
As the authors have only adjusted for the
number of natural teeth and self-perceived oral health status. If you can not adjust for comorbidity and general health status you have to explicitly address this as a limitation of the study.

L167: modify "
’ow-middle’ please.

Author Response

Dear reviewer

From author

Reviewer 2 Report

Comments and Suggestions for Authors

The authors have addressed the comments, and I am pleased with the revisions.

Author Response

Dear reviewer

From author

Reviewer 3 Report

Comments and Suggestions for Authors

Additional comments

1. Line 219-224, the OR presented in the text are from “adjusted model”, not from “the unadjusted group”. Also the OR show increased odds not “lower odds”. Please correct the text.

2. In Table 2/3, taking sex and unadjusted model as an example, why are there two lines of OR, 0.425, 95% CI 0.366-0.493 and 1.387, 95% CI,1.162-1.655? What does the first line of OR represent?

3. Line 232, should the OR be 0.999, 95% CI 0.999-0.999? Please also verify the OR presented in this paragraph.

4. Please address my previous comments regarding the interaction term. There is no description about it in the Statistic analysis (how it's modeled, etc.). In addition, ORs of those interaction terms are not presented in the Results section but are shown in the Abstract (line 20-22).

5. I don’t think the authors addressed my previous comment No. 5. For example, Of the studies of No. 13 and 35, neither study included sample from Taiwan and examined the association between dietary Ca intake and chewing difficulty (line 284-285).

Author Response

Dear reviewer

From author

Round 3

Reviewer 1 Report

Comments and Suggestions for Authors

I want to thank the authors for incorporating and addressing my comments. I have no further comments to add. 

Reviewer 3 Report

Comments and Suggestions for Authors

Minor comment

Line 295, Taiwan should be “Thailand”.